# Differences and Similarities between Front-of-Pack Nutrition Labels in Europe: A Comparison of Functional and Visual Aspects

**DOI:** 10.3390/nu11030626

**Published:** 2019-03-14

**Authors:** Daphne L. M. van der Bend, Lauren Lissner

**Affiliations:** 1Department of Epidemiology and Biostatistics, VU Medical Centre, Amsterdam Public Health Research Institute, 1081 HV Amsterdam, The Netherlands; 2Choices International Foundation, PO Box 10218, 2501 HE The Hague, The Netherlands; 3Department of Public Health and Community Medicine, Institute of Medicine, Sahlgrenska Academy, University of Gothenburg, SE-405 30 Gothenburg, Sweden; lauren.lissner@gu.se

**Keywords:** front-of-pack nutrition label, nutrient profiling, comparison, methodological approach, funnel model

## Abstract

Many different front-of-pack (FOP) nutrition labels have been introduced worldwide. To continue the debate on the most effective FOP labels for increased consumer health, full comprehension of their visual and functional features is relevant. This paper compares and provides an overview of all FOP labels currently in practice or in preparation in Europe, by means of the visually oriented Funnel Model. The Funnel Models were completed in collaboration with the respective FOP labelling initiatives. In total, six positive FOP labels, two mixed FOP labels and one negative FOP label were compared. There are multiple similarities and differences between the FOP labels, with each FOP label being characterised by a unique set of criteria and methodological approach. This Funnel Model comparison provides the knowledge to ultimately find more common ground for all stakeholders involved in the FOP labelling debate. Importantly, implementation and evaluation activities carried out by FOP labelling organisations are crucial success factors for FOP labels in practice. We conclude that more attention should be paid to methodological differences between FOP labels and recommend that the current comparison is expanded to a global level and periodically updated, as the variety of FOP labels in the global marketplace is changing constantly.

## 1. Introduction

Front-of-pack (FOP) nutrition labels are designed to simplify nutritional information presented on-pack to help consumers make healthier food choices, and stimulate healthy product reformulation. The World Health Organisation (WHO) recommends FOP labelling as a policy tool to tackle the global epidemic of obesity and diet-related noncommunicable diseases [1,2]. Across Europe, many food manufacturers and retailers have started to use these different FOP labels on their products, as many countries have introduced or started to introduce their own FOP labelling scheme [3,4,5].

There is an ongoing debate on what system is most effective in translating complex nutritional information for the consumer and many studies have been conducted on this topic [4,5,6]. Internationally, there is no agreement on mandatory FOP labelling, or on the specific formats or methodologies to be used in FOP labelling systems [1]. However, the Codex Alimentarius Commission has recently agreed to develop guiding principles for the provision of simplified nutrition information to consumers, enabling them to choose healthier food options through front-of-pack nutrition labelling [7].

While many studies conducted on FOP labels focus on consumer understanding, interpretation, appreciation and dietary behaviors, little research has focused on the underlying criteria and methodologies used in FOP labelling systems. Given the emergence of new FOP labels, in-depth insight into the different aspects that characterise existing FOP labelling systems is essential. These insights will create transparency and provide common knowledge on various aspects of different FOP labelling schemes.

During the Fourth European Logo Round Table in Copenhagen in January 2018, organised by the Choices International Foundation and attended by representatives of several European FOP labelling initiatives and WHO Europe, the need arose to develop an overview of the characteristics of all FOP labels in Europe. The idea was that this overview would be based on the ‘Funnel Model’, introduced in 2014 by van der Bend et al [8]. The model summarises and compares features of different FOP labelling schemes at a glance.

The aim of this paper is to update and apply the Funnel Model to FOP labels currently existing or to be implemented in Europe, making it possible to compare and critically discuss their functional and visual characteristics. This will lead to an increased understanding of their use, criteria and methodology, supporting relevant stakeholders, including scientists, policy-makers and industry, in their FOP labelling debate.

## 2. Materials and Methods

The Funnel Model, shown in Figure 1, distinguishes the following aspects of FOP labels (from top to bottom): (Dis-)qualifying components, Reference unit, Measurement method, Coverage, Methodological approach, Purpose, Driver, Directivity, Tone of voice and Utilization (Table 1). A combination of these aspects together forms the fundaments of a FOP labelling system, which is generally tested or validated before actual implementation. The aspects described in the Funnel Model are grouped into three different sections; i.e., the ‘Component’, ‘Methodology’ and ‘Expression’ sections. The latter two refer to the used methodology and how the label manifests itself in practice, respectively.

This paper focuses on all (semi-)directive FOP labels currently in practice or in preparation in Europe, with a national implementation, i.e., systems used by specific retailers were not included in this comparison. We have focused on (semi)directive labels specifically, as these are primarily subject to the current FOP labelling debate. Funnel Models were developed for the Keyhole label, the international Choices logo, Vim co jim (Czech Choices label), the Finnish Heart Symbol, the Croatian Healthy Living (HL) mark, the Slovenian Protective Food (PF) symbol, Multiple Traffic Light (MTL), Nutriscore label and the Israeli Warning label. The Israeli Warning label, which is yet to be implemented, is clearly distinctive from the other FOP labels, communicating a ‘negative’ health message, and therefore considered valuable to include in this European comparison. Since the Evolved Nutrition Label (ENL) is no longer under active development, we do not elaborate on this FOP label in our results but chose to include its Funnel Model in Appendix A for comparison.

Product criteria documents of the FOP labelling systems were used to complete the Funnel Models. Subsequently, each main driver, upon request, approved that the Funnel Model was filled in correctly.

## 3. Results

In the following sections, we will illustrate the use of the Funnel Model by highlighting two positive labels (Keyhole, Choices International), two mixed labels (Nutriscore and Multiple Traffic Light) and a negative label (Israeli Warning Label). For reasons of conciseness, we chose to only highlight two of the six positive labels. Appendix A show the Funnel Models of the other FOP labels.

### 3.1. Application of the Funnel Model

#### 3.1.1. Keyhole

The Keyhole label is the longest-standing FOP label in Europe. In 1989, the voluntary, European Union (E.U.)-notified Keyhole label was introduced in Sweden, originally by the Swedish retailer ICA Gruppen, and subsequently it was adopted by the Swedish National Food Agency. Since then, it has developed itself as a common Nordic label for healthier foods, when it was introduced subsequently by other members of the Nordic country cooperative (Denmark, Norway, Iceland), and in Lithuania and Macedonia (see Figure 2 for the Keyhole Funnel Model) [9]. The Keyhole label is a positive and directive label, aiming to help consumers to choose healthier food products within a product category, i.e., by using food-category-specific criteria, but also to stimulate healthy product reformulation. The Keyhole criteria do not apply to all products; hedonic products, such as sweets or snacks, have been excluded [9]. The Keyhole criteria are based on threshold values and expressed per 100 g/100 mL, per serving and in energy%, and they include both qualifying and disqualifying components [10]. Energy is included as both a disqualifying and qualifying component. Food additives or novel food with sweetening properties are specifically mentioned as disqualifying components.

#### 3.1.2. Choices International Foundation

The Choices International Foundation was founded in 2007, originally as an industry initiative, and has since then developed into a global platform for collaboration with industry, independent scientists, non-governmental organisations (NGOs) and health authorities, aiming to stimulate healthier food choices, and product reformulation. Because the Choices International criteria are based on a global analysis on nutrition recommendations, dietary guidelines and food composition databases, and therefore aim for global coverage, as illustrated in the map, Figure 3. In 2018, the Choices Board was reorganised and industry members resigned as board members to take seats within the Industry Support Group, advising the Board on long-term strategy and implementation, but not on updating criteria [11]. As a result, in the Funnel Model Choices is now highlighted as NGO instead of Commercial. Like the Keyhole label, the voluntary Choices label is a directive label conveying a positive health message (see Figure 3 for the Choices Funnel Model). Choices members currently include the Czech Republic, Nigeria and Argentina [12]. The Choices logo in The Netherlands was launched in 2006 and received national and E.U. approval in 2013, but it has recently been terminated [13]. Additionally, several European, Asian and African countries use the Choices criteria as a basis for their national healthier choices labels [12]. The Choices criteria are food-category specific. In contrast to Keyhole, the Choices criteria have been developed for all types of foods, including hedonic products, such as snacks, sweets and soft drinks. Furthermore, they include both qualifying and disqualifying components, for which specific thresholds have been developed, i.e., minimum values for fiber, and maximum values for energy, saturated fat (SFA), trans fat (TFA), sodium, total sugars and added sugars, respectively. The main difference with the Czech ‘Vim, co jim’ label is that ‘Vim, co jim’ has developed criteria for total salt instead of sodium. Nutrient thresholds are expressed per 100 g/100 mL or per serving (i.e., only for meals and snacks) [14,15]. Total sugar criteria have recently been added to provide a guideline for countries that do not have sufficient data on added sugar levels available [16].

#### 3.1.3. Nutriscore

In 2017, the voluntary Nutriscore FOP labelling scheme was initiated in France, and it was recently also approved to be used in Belgium, Spain and Portugal by their respective Ministries of Health [17,18,19,20]. The main purposes of the Nutriscore label are to help consumers make healthier choices and to stimulate product reformulation towards healthier product compositions (see Figure 4 for the Nutriscore Funnel Model) [17]. In contrast to the Keyhole and Choices label, the Nutriscore label conveys a mixed message as it displays five boxes with colors ranging from dark green to dark red, with letters to grade foods according to their overall nutritional quality; from A for products with the ‘best nutritional quality’ to E for the products with the ‘least good nutritional quality’ [17]. Because Nutriscore provides a summary indicator for each food along the continuum from healthy to unhealthy, it is considered neither positive nor negative. Therefore, it is rather viewed as a mixed scheme. As only colors and letters are used to indicate the healthfulness of a food product and no factual information is presented, such as specific nutrient levels or percentages of daily intake, Nutriscore is considered a directive FOP label, like the Keyhole and Choices label. The criteria are based on a scoring as well as a threshold method, covering both qualifying and disqualifying components, and they are expressed per 100 g/100 mL. First, a total score, ranging from −15 to +40, is calculated, consisting of two dimensions: positive points (0–10) are assigned to disqualifying components, such as SFA or sodium, and negative points (0–5) are assigned to each qualifying component, such as protein or fiber. Which box (A–E) will be magnified depends on specific lower and upper bounds that are defined for each of the five boxes. The Nutriscore is based on one set of criteria for all pre-packaged foods with a mandatory nutritional declaration in accordance with Regulation (E.U.) No. 1169/2011, although criteria modifications have been made specifically for cheeses, fats and non-alcoholic drinks, because the score of these products would not be in line with dietary recommendations [17].

#### 3.1.4. Multiple Traffic Light Scheme

The voluntary, mixed Multiple Traffic Light (MTL) scheme was launched in 2013 by the U.K. Department of Health (DH), primarily aiming to help consumers make healthier food choices (see Figure 5 for the MTL Funnel Model) [21,22]. The MTL scheme complies with the U.K. Health Ministers’ Recommendation on the use of color coding and with the E.U. Regulation (No. 1169/2011) on the provision of food information to consumers (E.U. FIC) [21]. In contrast to all other FOP labels in this comparison, the MTL is a semi-directive FOP label, as it combines green, amber and red color-coding with percentage Reference Intakes (RIs, formerly known as Guideline Daily Amounts) to display the amount of energy, total fat, saturated fat, total sugar and salt in foods and drinks [21]. In line with the E.U. FIC, the MTL should be provided in either one of the following two formats: energy alone or energy plus total fat, saturates, total sugars and salt (‘energy + 4’). On-pack, reference bases are provided per 100 g/mL only, per 100 g/mL and per portion, or per portion only (applies only for ‘energy + 4’). When the latter is applied, energy must be provided per 100 g/mL in addition to per portion [21]. The nutrients (i.e., not energy) in the MTL are colored based on specified upper and lower bounds per 100 g/mL, which are developed for green, amber and red colors and are different for food and drink products. If portion/serving sizes of foods or drinks are larger than 100 g or 150 mL, respectively, portion size criteria apply for the color red specifically. The MTL is considered to be an across-the-board system as it applies generic criteria to foods and drinks and does not apply criteria for specific food or drink categories [21].

#### 3.1.5. Israeli Warning Label

The Israeli Warning Label, approved by the Israeli parliament’s Labor, Welfare and Health Committee in 2017, is a mandatory FOP label with a negative tone of voice and a directive message (see Figure 6 for the Warning Label Funnel Model). With this new label, the Israeli Ministry of Health aims to allow consumers to have an informed choice of foods, and promote product reformulation. In contrast to the Keyhole, Choices, Nutriscore and MTL label, the Israeli Warning Label is mandatory, and it will be displayed on all products exceeding certain threshold levels of disqualifying nutrients, i.e., SFA, sodium and total sugar, indicated by ‘High saturated fat level’, ‘High sodium level’ and ‘High sugar level’, respectively [23]. Different criteria for solid and liquid products have been developed, but no criteria have been specified for solid or liquid subcategories [23]. Food products that are not impacted by the new Israeli labelling regulation include all products not considered to be pre-packaged (i.e., fruits, vegetables, meats, fresh eggs and prepared foods purchased at food service establishments), and products such as tea, coffee, yeast, spices and tabletop sweeteners [24]. It is expected that the first phase of the Israeli Warning Label will go into force in January 2020. From then, the 12-month transition phase will start, which will include a first set of requirements for the disqualifying components mentioned above. In the second (permanent) phase, starting from January 2021, the threshold levels defined in the first phase will become stricter [24]. Additionally, the Israeli government is developing a positive counterpart of the Warning label, which will have a green color [25].

### 3.2. Comparison of FOP Labels in Europe

#### 3.2.1. Comparison of Positive FOP Labels

The positive FOP labels mainly include disqualifying components in their product criteria. Compared to the other positive European FOP labels, the Slovenian Protective Food (PF) symbol includes a relatively large number of qualifying components in its product criteria. Of all positive FOP labels, the Finnish Heart Symbol includes the fewest components (Appendix A). The components that are being used in all positive FOP labelling systems are dietary fiber, SFA, total salt or sodium, and total sugar. Qualifying components that are used in at least two of the six positive FOP labelling systems are omega-3 fatty acids, fruit and vegetables, wholegrain, dietary fiber and legumes. Disqualifying components that are used in at least four of the six positive FOP labelling systems are energy, total fat, SFA, TFA, total sugar, added sugar, and total salt or sodium.

The positive FOP labels are relatively similar in that they are all directive, voluntary, aim to help consumers make informed choices and stimulate healthy product reformulation, are category specific and use a threshold method (Table 2). However, there are some differences in coverage; the International and Czech Choices label and the Croatian Healthy Living mark have developed criteria for all food products, while the Finnish Heart Symbol, Keyhole and Slovenian PF symbol have excluded some food categories, i.e., mostly discretionary foods, such as snacks, sugar-sweetened beverages or bread toppings. Furthermore, only the Keyhole and Croatian label are governed by governments; the other four positive FOP labels are NGO-driven.

#### 3.2.2. Comparison of Positive, Mixed and Negative FOP Labels

Similar to the positive labels, the mixed and negative FOP labels include disqualifying components in their criteria. However, they do not include any qualifying components, except for Nutriscore (Appendix A). Also, the Nutriscore criteria include protein, which is not present in the criteria of any of the positive labels. Of all disqualifying components, SFA, total sugar and total sodium or salt are included in the criteria of all mixed and negative labels. While added sugar is included in the criteria of almost all positive FOP labels, none of the mixed or negative labels have included this component. Other components that are included in the criteria of (some) positive labels but in none of the mixed or negative labels are TFA, cholesterol, plant sterols/stanols, added fats, artificial sweeteners, added sodium, additives, alcohol and free fatty acids. The Israeli label includes the fewest components of all positive, mixed and negative labels (Appendix A).

Some characteristics are found to be similar across positive, mixed and negative FOP labels. They all include disqualifying components related to the levels of sodium, sugar and fats in their product criteria, they use 100 g/100 mL as reference unit, they qualify products based on threshold values and they aim to help consumers make healthier choices (Table 2). The most important difference between the positive labels and the mixed or negative labels is that the positive FOP labels all use a category-specific approach, i.e., applying different criteria for different food categories, while the mixed and negative FOP labels generally use one set of criteria for all food categories, with sometimes a few exceptions regarding specific food categories, e.g., see Nutriscore. Furthermore, Nutriscore is the only FOP label that qualifies products based on both scoring and threshold values. Product reformulation is a primary aim for most of the FOP labels, except for the Multiple Traffic Light (Table 2). Also, the MTL is the only semi-directive label, as it shows absolute nutrient values and also uses simple color schemes. The majority of the FOP labels (i.e., MTL, Nutriscore, Choices, Vim co jim and the Croatian HL mark) have developed criteria covering all types of products. Most of the FOP labels are governed by governments (i.e., Keyhole, Croatian HL mark, MTL, Nutriscore and Israeli Warning Label). None of the labelling systems are governed by industry. Additionally, at present, the Israeli Warning Label is the only mandatory FOP label (Table 2).

## 4. Discussion

This paper describes all FOP labels currently in practice or in preparation in Europe using the Funnel Model as a tool. In total, nine FOP labels, including six positive, two mixed and one negative FOP label were compared by highlighting their features in the Funnel Model. Of all nine FOP labels, the total number of (dis)qualifying components used ranges from 3 components (Israeli Warning Label) to 18 components (Slovenian Protective Food (PF) symbol). The Multiple Traffic Light (MTL) and Israeli Warning Label are the only FOP labels solely including disqualifying components in their criteria. Characteristics that are part of all FOP label schemes include: type of component (i.e., disqualifying components), reference unit (i.e., 100 g/ 100 mL), measurement method (i.e., threshold) and main purpose (i.e., aiming to help consumers make healthier food choices).

The main distinction between the positive FOP labels and the other label types is the methodological approach; all positive FOP labels use specific criteria for different food categories, whereas the mixed and negative FOP labels use an across-the-board approach and for example only distinguish solid and liquid food. Nutriscore applies some modifications to the criteria for cheese, fats and beverages because of their distinctive nutritional composition. Previous research suggests that criteria specifically tailored to product categories take into account the variability of the nutritional composition of foods that are part of the diet, and therefore may be better able to stimulate product reformulation than across-the-board systems [26,27]. However, at present, there is no evidence on the optimal number of categories to be used [26]. While a higher number of food categories take into account variable compositions of foods across food categories, a smaller number of categories may make it easier to define and apply in regulations, and classifying foods may be less accompanied by subjectivity [28]. In short, further study into the effectiveness of the across-the-board and food-category-specific approaches is warranted.

Total sugar, SFA and total sodium or salt are disqualifying components used in all positive, mixed and negative FOP labelling systems in Europe. A recent systematic review that summarises all global nutrient profile models with applications in government-led nutrition policies (i.e., primarily school food standards, FOP nutrition labelling and restriction of the marketing of foods and beverages to children) indeed shows that SFA, total sugar and sodium were included most frequently in these models [28]. Data on these nutrients are usually readily available in food composition databases, making it easier to evaluate whether foods meet FOP labelling requirements. However, it is increasingly being argued that more emphasis should be put on free or added sugars instead of total sugars in dietary guidelines [29]. Additionally, among all nutrient profile models, the three qualifying components included most frequently were ‘fruits, vegetables, nuts and legumes’ (i.e., ingredients of plant origin), fiber and protein [28]. This is largely in line with our findings, as fruit, vegetables, legumes and fiber are among the qualifying components included in most FOP labels. However, protein was only included in the Nutriscore criteria.

The MTL label is the only label studied here not specifically stimulating product reformulation. Additionally, the MTL label is distinct from the other FOP labels in that it is semi-directive, combining colors with nutrient values; the other FOP labels are all directive, providing a summary indicator to describe the nutritional quality or ‘healthiness’ of a food product. Compared to non-directive FOP labels, semi-directive and directive FOP labels may be better able to increase consumers’ understanding of nutritional information on a pack. In particular, simpler or more directive FOP labels may be most effective in helping consumers make fast healthy choices, especially in settings that are characterised by situations involving quick decision-making, e.g., supermarkets [30].

All FOP labels described here are voluntary, except for the Israeli Warning Label. Research on differences in the effectiveness of voluntary and mandatory FOP labels on consumer behavior and product reformulation is scarce, and this should therefore be explored more thoroughly, especially now that mandatory labels, such as the Israeli Warning label, will be introduced on the market. However, preliminary evidence from outside Europe indicates that there is much opposition from industry against mandatory nutrition labelling policies [31]. On the other hand, market uptake of voluntary FOP labels is relatively slow [32], and several factors, including moral motives (i.e., responsibility) and instrumental motives (i.e., profit) may play a large role in food firms’ decisions to reformulate products in order to obtain a FOP label [33]. Also, previous research has shown that, when not all packages are labelled, this may bias consumers’ perception towards labelled products that are equally or less healthy than products without a label [34]. Ideally, consumers would need to know whether a product is either not eligible for or does not participate in the FOP labelling scheme.

Importantly, most FOP labels are currently supported by governments, followed by NGOs. In contrast, the Evolved Nutrition Label (ENL) was supported by industry, but it received criticism for providing nutrition information per serving instead of per 100 g, as this would result in potentially misleading color codes [35]. Theoretically, portion sizes can be changed in order to meet certain criteria, while the ingredients of a food product are still the same. A ‘per serving’ approach can introduce several difficulties; not only is it challenging to compare products’ compositions on the basis of their serving size, but also serving sizes and consumption patterns are individual matters that cannot easily be standardized [36]. However, to obtain a complete picture on reference units, more research should be conducted on the application and effectiveness of using serving sizes in FOP labelling criteria. The Keyhole, Choices and MTL labels also use serving sizes in their criteria, but only for specific food categories. The MTL takes serving size into account when the serving size of the product is more than 100 g or 150 mL, and Keyhole and Choices only apply portion size criteria for meals [10,16,21].

Since 2014, newly added to the Funnel Model comparison are Vim co Jim, the Croatian Healthy Living mark, Nutriscore and the Israeli Warning Label. Changes in the earlier-published Funnel Models of Choices International, Keyhole, Finnish Heart Symbol, Slovenian PF symbol and Multiple Traffic Light may be the result of changes in their criteria, or the organisational structure of their main driver. For example, Choices’ internal structure has changed recently as industry no longer is represented in the Choices board, which resulted in a change of the main driver in the Funnel Model from industry to NGO. Other changes are the result of further refinement of the previous Funnel Models by using input from the main drivers of the FOP labels. Involving them in the verification of the Funnel Models is a new element of the Funnel Model comparison, which may have contributed to the reliability of this study.

There are a few important points to consider with regard to this paper. First of all, we have mainly discussed FOP labels currently in practice or in preparation in Europe. A recommended next step would be to include FOP labels in use worldwide and structurally compare these using the Funnel Model as a tool. Secondly, one should keep in mind that, when comparing Funnel Models of different FOP labelling systems, minor differences may still have large implications in practice. For example, when comparing the Funnel Models of the ENL and MTL, these systems seem to be quite similar, while the use of portion sizes in combination with involvement of industry has led to criticism towards the ENL among several stakeholders. As the ENL is currently under discussion, it is not yet clear whether and how the label will continue. Thirdly, the Funnel Model does not focus on coherence or synergy with additional implementation activities carried out by FOP labelling initiatives, while such activities could be critical success factors for the use of FOP labels in practice. This may include consumer education, i.e., FOP labels need to be explained to consumers when implemented. Such education may take place for many years in order to build the label’s credibility, recognition and understandability. In addition, a FOP label can be integrated in overall consumer education to promote healthier dietary behaviors. For example, in Norway, education material was developed for primary schools that includes information about healthy nutrition and the Keyhole label [37]. Furthermore, implementation may include national adaptations of the criteria based on national differences (e.g., cultural eating customs, national food supply, national nutrition policy agenda and legislation). This issue is particularly relevant for FOP labels that are (to be) implemented in more than one country, i.e., the MTL, Keyhole, Choices logo, and Nutriscore. For example, the Choices International Foundation pays special attention to national adaptations of the international Choices criteria by guiding national labelling initiatives in the development of country-specific criteria [38]. In contrast, Keyhole only allows unmodified use of the criteria. Finally, besides monitoring the impact of implementation activities, one could think of relevant evaluation activities of specific aspects of the labelling system itself, such as periodical revisions of the product criteria, to make product reformulation feasible and realistic for food companies (i.e., Keyhole, the Finnish Heart Symbol, Choices logo [9,38,39]), or periodical market surveys to ensure correct use of the logo on-pack, which has been carried out yearly by the Dutch Choices initiative [40]. All these additional activities, initiated by FOP labelling initiatives, may largely impact FOP labels’ effectiveness in practice, and are therefore highly relevant to take into account when evaluating their visual and functional differences. It is therefore recommended to include these aspects in further worldwide comparisons.

Globally, there is much debate about which FOP label is most effective for improving dietary behaviors of consumers. The results of this paper have shown that each FOP label is unique in several respects, suggesting that it may be difficult to define what particular combination of FOP labelling characteristics is optimal, especially since many additional contextual factors are involved [41]. The Funnel Model provides a basic framework that can be utilized to link the methodologies used in different FOP labelling systems to outcomes of effect studies, ultimately aiming to create a better understanding of why FOP labels impact consumers differently. Yet, the actual impact of FOP labels on obesity and diet-related non-communicable disease rates remains to be determined, as many studies investigating potential health effects have used modelling approaches, and did not analyse real observational data [42]. At least long-term and large-scale research on the effect of FOP labels in real-life contexts is recommended, i.e., measuring purchasing behavior in supermarkets instead of conducting surveys or experiments in labs or virtual environments. An example of such research is the study conducted by Smed et al., as part of the CLYMBOL project; preliminary results show that consumers in The Netherlands increased their purchases of products with the Choices logo [43]. Such empirical research is particularly valuable as the effectiveness of FOP labels may not simply be determined by their visual and functional aspects, but also by additional implementation activities carried out by FOP labelling initiatives, in coherence with other public health measures, by stakeholder support and by independent evaluation studies. In addition, as voluntary FOP labelling schemes only cover part of the market and therefore may induce bias among consumers with regard to identifying healthy products, conducting research on mandatory (semi-)directive FOP labels, e.g., the Israeli Warning Label, is essential, and should preferably be carried out from the start of implementation. Furthermore, consumer understanding of FOP labels may be dependent on sociocultural context and differs across cultures or countries [44]. Research on this topic is scarce, although a recent analysis comparing objective consumer understanding of five FOP labels in 12 different countries has suggested that Nutriscore may be better understood by consumers than other FOP labels conveying mixed or negative health messages [45]. However, no conclusions can be drawn on positive FOP labels, which were not included in that analysis.

Importantly, to date, many studies on FOP labelling have focused on consumer understanding, while only limited research has been conducted on the criteria or methodological approaches used in FOP labelling systems. The current paper adds to the 2014 Funnel Model paper by describing and discussing the underlying methodologies used in positive, mixed and negative FOP labelling systems as well as their potential implications in practice, while acknowledging the importance of additional evaluation and implementation activities carried out by FOP labelling initiatives. We recommend that more research is conducted on the rationale behind and application (of the criteria) of FOP labelling systems in practice, e.g., by exploring types of qualifying versus disqualifying components, scoring versus threshold methods, across-the-board versus food-category-specific approaches, nutrient level per 100 g versus per serving units, or mandatory versus voluntary use. It is essential that each of these features are studied repeatedly, not only in relation to consumer understanding, but also product reformulation, applicability and ease of use in practice, in different cultural settings. The results of such studies may provide fundaments for a FOP labelling system to be implemented in multiple settings.

To conclude, this paper provides a comprehensive and updated overview of the different functional and visual features of all FOP labels currently in practice or in preparation in Europe. Ultimately, full comprehension and awareness of the different characteristics of FOP labels by all involved stakeholders, outlined against the cultural contexts in which each of these labels thrives, can bring the FOP labelling debate to the next level and help interpret the outcomes of effect studies. This is valuable, especially since the variety of FOP labels in the marketplace is dynamic and constantly changing as many countries are currently considering introducing FOP labels as national health tools [1,5]. This emphasises the need to further expand and continue updating this present overview of existing FOP labelling schemes and stay informed on the global FOP labelling developments over time.

## Figures and Tables

**Figure 1 nutrients-11-00626-f001:**
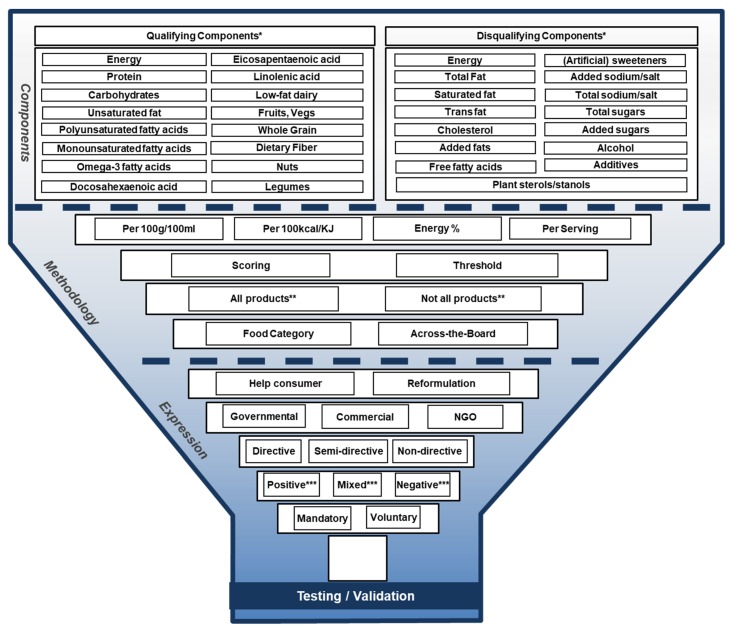
The Funnel Model describing functional and visual aspects of front-of-pack (FOP) labels in Europe. * ‘Ingredients’ in the 2014 Funnel Model was replaced by ‘components’. ** Coverage was added to the 2014 Funnel Model as an additional FOP label characteristic. *** Tone of voice was added to the 2014 Funnel Model as an additional FOP label characteristic.

**Figure 2 nutrients-11-00626-f002:**
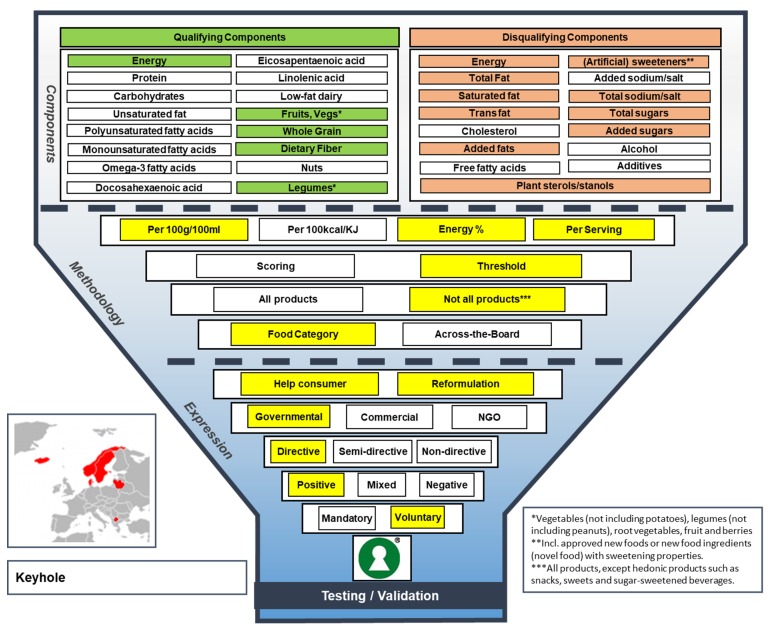
The Funnel Model describing functional and visual aspects of the Keyhole label.

**Figure 3 nutrients-11-00626-f003:**
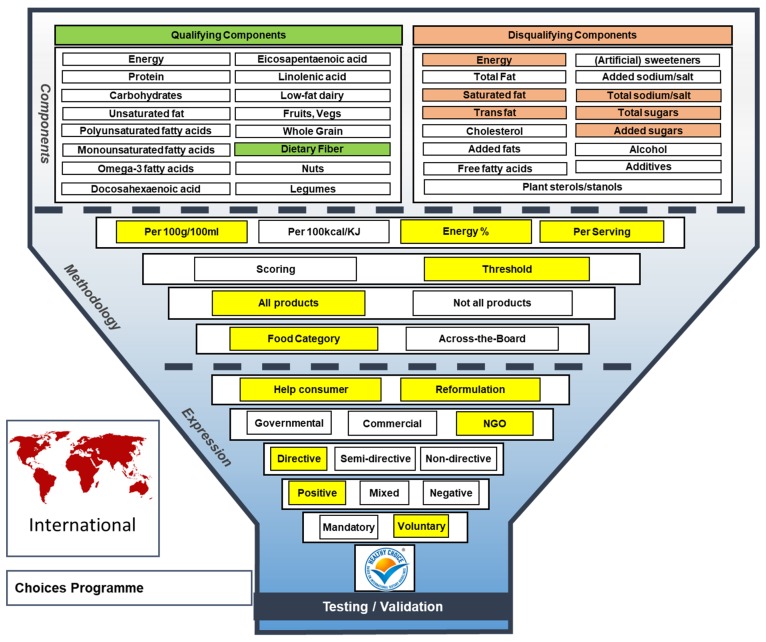
The Funnel Model describing functional and visual aspects of the international Choices label.

**Figure 4 nutrients-11-00626-f004:**
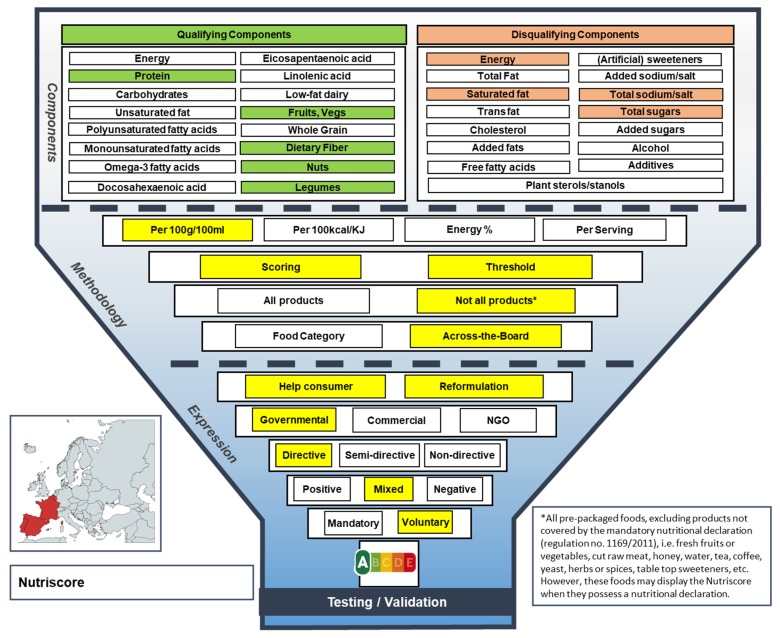
The Funnel Model describing functional and visual aspects of the Nutriscore label.

**Figure 5 nutrients-11-00626-f005:**
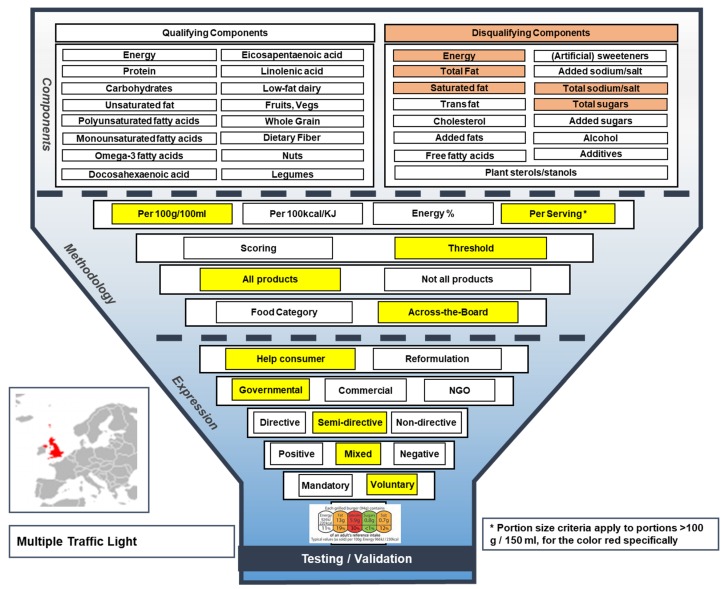
The Funnel Model describing functional and visual aspects of the Multiple Traffic Light scheme.

**Figure 6 nutrients-11-00626-f006:**
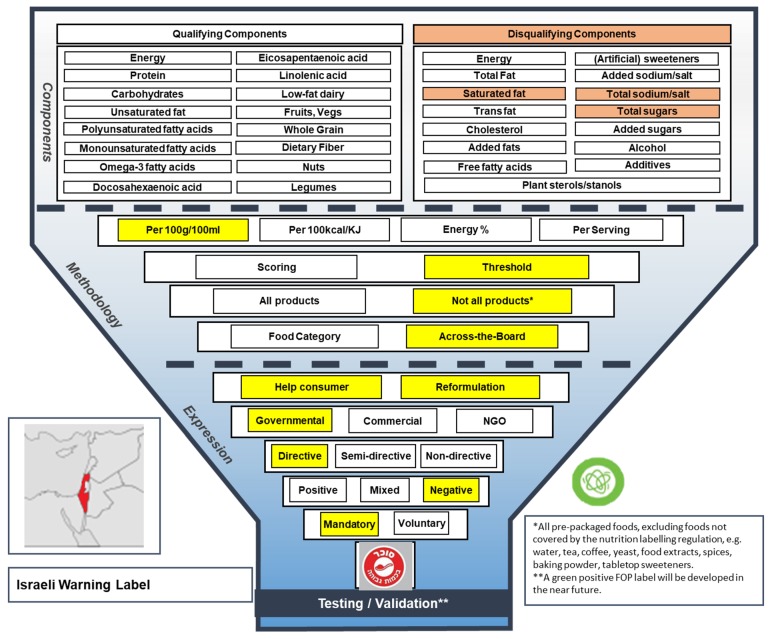
The Funnel Model describing functional and visual aspects of the Israeli Warning Label.

**Table 1 nutrients-11-00626-t001:** A summary of indicators that are used in the Funnel Model to describe aspects of front-of-pack (FOP) labels.

Indicator	Explanation
Components	Product criteria of FOP labels may take into account qualifying components, i.e., components in a food product beneficial for health, and/or disqualifying components, i.e., components in a food product with a negative impact on health.
Reference unit	Product criteria of FOP labels may be expressed per 100 g/100 mL, per 100 kcal/KJ, in Energy% and/or per serving, amongst others.
Measurement method	Compliance of foods with the FOP label’s product criteria may be determined on the basis of calculated scores and/or threshold values.
Coverage	Product criteria of FOP labels are either developed for a selection of food categories, or they cover all food categories at once. ‘All food categories’ includes at least all pre-packaged foods, but does not include specific products, such as infant formula, alcoholic beverages and food supplements.
Methodological approach	When FOP labelling systems make use of the same set of criteria for all or most food categories, they use an across-the-board approach. When different criteria have been developed for different food categories, a food-category-specific approach is used. We do not consider liquid versus solid foods to be food-category specific, as the composition of food categories within these groups can still be very variable.
Purpose	The primary aim of FOP labels may be, for example, to inform consumers about the nutritional contribution a food product makes to the diet, help consumers identify healthy foods and/or to stimulate product reformulation by the food industry. FOP labels may have several purposes.
Driver	This refers to the driving force behind a FOP label (at the time of the writing of this article); a driver may be governmental, commercial or be part of a non-governmental organisation (NGO).
Directivity	This specifies to what degree the FOP label leaves interpretation of ‘healthiness’ of a product to the consumer. Non-directive FOP labels only present factual nutrient information, semi-directive FOP labels combine factual information with easy-to-interpret visuals (e.g., color coding), and directive FOP labels merely summarise the ‘healthiness’ of a product without displaying any nutritional information.
Tone of voice	A FOP label may convey a positive (‘healthy’), mixed (mixture of ‘healthy’ or ‘unhealthy’) or negative (‘unhealthy’) health message.
Utilization	In case of voluntary use, food firms may choose whether or not to use the FOP label on-pack. When a FOP label is mandatory, often determined by national regulations or legislation, food firms are forced to use the label.

**Table 2 nutrients-11-00626-t002:** Characteristics of positive, mixed and negative FOP labels in Europe.

	Positive FOP Labels	Multiple Traffic Light	Nutriscore	Israeli Warning Label
**Components**	Qualifying, disqualifying	Disqualifying	Qualifying, disqualifying	Disqualifying
**Reference unit**	100 g/100 mL, 100 kcal/KJ, per serving, energy%	100 g/100 mL, per serving	100 g/100 mL	100 g/100 mL
**Measurement method**	Threshold	Threshold	Threshold, scoring	Threshold
**Purpose**	Help consumer, reformulation	Help consumer	Help consumer, reformulation	Help consumer, reformulation
**Methodological approach**	Category specific	Across-the-board	Across-the-board	Across-the-board
**Coverage**	Differs per label	All products	Not all products	Not all products
**Driver**	Governmental, NGO	Governmental	Governmental	Governmental
**Directivity**	Directive	Semi-directive	Directive	Directive
**Tone of voice**	Positive	Mixed	Mixed	Negative
**Utilization**	Voluntary	Voluntary	Voluntary	Mandatory

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
