# Peer review of "Differences and Similarities between Front-of-Pack Nutrition Labels in Europe: A Comparison of Functional and Visual Aspects"

_nutrients, 2019, doi:10.3390/nu11030626_

Round 1
Reviewer 1 Report
I acknowledge the edits made to the manuscript and would like to highlight the following outstanding issues:
- Unless the international Choices logo is used in Europe with unmodified criteria, it should not be discussed in the main manuscript (as it is neither in practice nor in preparation in Europe). Inclusion of the funnel model in the supplemental information would be acceptable.
- The UK MTL description should clarify that the portion criteria for the colour-coding apply only for the colour red.
- The image used for the ENL is the UK Multiple Traffic Light. This must be changed. Given that the website of the ENL has been taken down, it is worth considering complete removal of the ENL parts from the manuscript.
- The statement in ll. 486-490 sounds quite opinionated. I would recommend stating the added value in a more factual manner, similar to the addition made in lll. 460-463.
Author Response
Comments and Suggestions for Authors
I acknowledge the edits made to the manuscript and would like to highlight the following outstanding issues:
- Unless the international Choices logo is used in Europe with unmodified criteria, it should not be discussed in the main manuscript (as it is neither in practice nor in preparation in Europe). Inclusion of the funnel model in the supplemental information would be acceptable.
We do not fully agree. In our opinion, it is important to include the international Choices logo in the main text as it has a global coverage, incl. Europe (as stated in lines 124-126), serving as an inspiration for several positive FOP labels in Europe. Its criteria can be used in either unmodified of modified format by any European country (as referred to in 361-363); several European countries have tailored it to their national situation, including the Czech Republic and Croatia. Also, there are examples of non-European countries that have accepted the Choices criteria unaltered (i.e. Argentina), or developed their own criteria using the Choices criteria as an example (i.e. Malaysia, Thailand, Brunei). Moreover, the international Choices logo is currently included in several overviews, studies and discussions on FOP labelling that are highly relevant for Europe, for example in the CLYMBOL study; the studies included in the thesis by Vyth et al.; the recent report on FOP labelling by the Codex Alimentarius (2017), and the Joint Research Centre (JCR) report on available FOP systems which is currently in preparation. Additionally, Choices was asked by the Max Rubner-Institute, the German Federal Research Institute of Nutrition and Food (Prof. Dr. Pablo Steinberg) to contribute to their exploration of existing logos in preparation of a decision about FOP labelling in Germany. Altogether, we believe that it is relevant to include of the Choices logo in the main text.
- The UK MTL description should clarify that the portion criteria for the colour-coding apply only for the colour red.
We agree, and made changes in the text (see lines below), and in the Figure of the MTL Funnel Model footnote.
Line 188-189: ‘If portion/serving sizes of foods or drinks are larger than 100g or 150 ml respectively, specific portion size criteria apply for the colour red specifically.’
- The image used for the ENL is the UK Multiple Traffic Light. This must be changed. Given that the website of the ENL has been taken down, it is worth considering complete removal of the ENL parts from the manuscript.
We have changed the image for the ENL. Regarding the other remark about removing ENL parts from the manuscript, we do not fully agree with this. The reviewer is right that the website has been taken down, but ENL is actually still in practice at the moment; for example, it was printed on packages by Coca-Cola in the Netherlands, and in Bulgaria some of the ENL companies have started introduction as well. So, the ENL is indeed suspended, but as it has not completely disappeared from the market and is still under discussion, we have left the ENL Funnel Model in the Supplementary Material. Moreover, we believe that it is relevant and interesting to show the ENL Funnel Model for comparison and mention ENL in the discussion on different aspects of FOP labels in Europe, mainly with regard to the issues on portion size and involvement of industry. However, to highlight that it is unclear how ENL will continue and that it is under serious discussion at the moment, we have added the following line:
Line 349-350: ‘As the ENL is currently under discussion, it is not yet clear whether and how the label will continue.’
- The statement in ll. 486-490 sounds quite opinionated. I would recommend stating the added value in a more factual manner, similar to the addition made in lll. 460-463.
We agree with the reviewer that part of sentence 486-490 could be perceived as opinionated. Therefore, we made a small change to the sentence:
Line 402-406: ‘The current paper adds to the 2014 Funnel Model paper by describing and discussing the underlying methodologies used in positive, mixed and negative FOP labelling systems as well as their potential implications in practice, while acknowledging the importance of additional evaluation and implementation activities carried out by FOP labelling initiatives.’

Reviewer 2 Report
the responses look satisfactory.
Author Response
Comments and Suggestions for Authors
the responses look satisfactory.
Thank you, we think your comments have contributed to the quality of our paper.
This manuscript is a resubmission of an earlier submission. The following is a list of the peer review reports and author responses from that submission.
Round 1
Reviewer 1 Report
Nutrients 429112 – “differences and similarities between front-of-pack nutrition labels in Europe: a comparison of functional and visual aspects”
General comment: This study aimed at using the Funnel Model to describe and compare the front-of-pack (FOP) labels currently existing in Europe and Israel. The study provides a useful overview of the characteristics of reviewed labels and the manuscript is well-written. Below are some minor suggestions for the author(s)’ consideration to enhance the contribution of the manuscript.
Specific comments:
Lines 43-48 (or at the end of the paper): suggest adding a brief discussion of the relevance and importance of information provided in the funnel models. Specifically, the effectiveness of any FOP label depends on not only its nutrition science basis but also on how consumers understand and use the label. Here, both functional and visual aspects of a label can affect consumer understanding and responses. Therefore, the model information can provide a useful context to understand why various labels may have different effects on consumers. For example, it is expected that directive labels are easier to understand and use than semi- or non-directive labels because the former requires the least cognitive effort by consumers.
Lines 66-67: the sentences are not clear. Are there missing words?
Keyhole and Choices models: these models are different from those in van der Bend et al. (2014). If all the differences are due to changes discussed in Lines 286-294, then that discussion should be included in the respective model so readers can understand the origin of the differences.
Author Response
Comments and Suggestions for Authors
Nutrients 429112 – “differences and similarities between front-of-pack nutrition labels in Europe: a comparison of functional and visual aspects”
General comment: This study aimed at using the Funnel Model to describe and compare the front-of-pack (FOP) labels currently existing in Europe and Israel. The study provides a useful overview of the characteristics of reviewed labels and the manuscript is well-written. Below are some minor suggestions for the author(s)’ consideration to enhance the contribution of the manuscript.
Specific comments:
Lines 43-48 (or at the end of the paper): suggest adding a brief discussion of the relevance and importance of information provided in the funnel models. Specifically, the effectiveness of any FOP label depends on not only its nutrition science basis but also on how consumers understand and use the label. Here, both functional and visual aspects of a label can affect consumer understanding and responses. Therefore, the model information can provide a useful context to understand why various labels may have different effects on consumers. For example, it is expected that directive labels are easier to understand and use than semi- or non-directive labels because the former requires the least cognitive effort by consumers.
We agree with the reviewer that it is relevant to acknowledge the importance of linking the different FOP label features in the Funnel Models to effect studies and impact on the consumer behavior/understanding. We have added a sentence about this in the discussion section.
Lines 375-378: ‘The Funnel Model provides a basic framework that can be utilized to link the methodologies used in different FOP labelling systems to outcomes of effect studies, ultimately aiming to create a better understanding of why FOP labels impact consumers differently.’
Lines 66-67: the sentences are not clear. Are there missing words?
Rightly noted by the reviewer, we have changed the sentence.
Line 85: ‘Product criteria documents of the FOP labelling systems were used to complete the Funnel Models.’
Keyhole and Choices models: these models are different from those in van der Bend et al. (2014). If all the differences are due to changes discussed in Lines 286-294, then that discussion should be included in the respective model so readers can understand the origin of the differences.
It is not necessarily the goal of this paper to discuss what has changed since 2014, as the main aim of our paper is to focus on the characteristics of the FOP labels currently in practice or in preparation. Therefore we have not included a further discussion on this in the Funnel Models.
Reviewer 2 Report
The study by van der Bend and Lissner presents a comparison of the characteristics of different front-of-pack (FOP) labelling schemes from Europe and Israel, using a self-developed graphical model. While the study is timely given the ongoing debate in the EU, the proposed funnel model only allows a standardised description of FOP scheme characteristics; care should be taken not to venture too far beyond the consequentially limited scope.
Major comments
While the funnel model allows for a standardised recording of a range of FOP scheme characteristics, it is not clear from the manuscript what the true value of this assessment is, and who would be the main users of the information obtained. The Codex Alimentarius Commission has recently produced a global inventory of FOP schemes and their characteristics - what does your study add to the Codex database?
What is the reasoning for using the funnel shape? From a semantic angle, the funnel model may be read from top to bottom as a specific combination of criteria resulting in one specific FOP scheme. Obviously this is not the case, thus the authors may want to reconsider the shape of the graphic.
Why is the model divided into three sections, and what links the elements within the middle and the bottom sections? Furthermore, what is the meaning of the part saying "Testing/Validation"? Currently it does not seem to contain any information?
There is no clear rationale for the inclusion or exclusion of specific FOP systems. The authors claim that "all FOP labels currently existing in Europe and Israel" were compared, yet:
there is no mention of the most widespread FOP scheme called Reference Intakes (RI);
the Israeli FOP system is not yet in use;
the inclusion of the Czech Vim co jim logo creates a bias towards Choices (the Croatian Healthy Living logo was also inspired by Choices);
other FOP systems such as the traffic light schemes used by the retailers Eroski in Spain, Continente in Portugal, Rimi in Estonia are not considered.
In light of the authors' statement that no positive logo was included in the 12-country study by Egnell et al., efforts should be made to show a more comprehensive picture of different FOP scheme types in the present study. In my view, the Vim co jim logo should be removed from the analysis. Furthermore, it would seem prudent to give more space to the UK Multiple Traffic Light-RI hybrid scheme in the main text, including the corresponding funnel model.
Another issue with the funnel model is evidenced by the supplemental figure S3. Apart from the driver (governmental vs commercial), all other boxes are coloured in the same, suggesting the two systems are essentially the same. However, research suggests that consumers try to avoid MTL with lots of reds and remain rather indifferent to MTL without any reds. Consequently, seemingly identical systems may yield very different responses in real life, a fact not or only poorly reflected in the funnel model. It is also noteworthy here that the visuals used for the MTL and the ENL are the same in the funnel models, which I would strongly advise against. Better to take the MTL graphic from the official UK guidance at https://www.gov.uk/government/publications/front-of-pack-nutrition-labelling-guidance.
The authors point out that each FOP scheme has its own unique features (l. 20) and at the same time they lump all positive logos together in Table 2, thus leaving the reader unclear as to which reference units and purposes apply to which scheme, or who is the driver behind which scheme. This lack of clarity then hampers understanding of the field "Components", where both options listed apply to all. Overall, Table 2 seems of little use to me – suggest deletion.
Since the present study is all about the application of the funnel model as a comparative tool, the various supposed errors and inaccuracies noted in the model graphics may also be considered a major issue. For example, according to my information the Choices logo has E% criteria for "All other" non-basic foods for SFA, TFA, and added sugars. The Croatian Healthy Living mark does not have criteria for added sodium but for added salt, and it includes disqualifying criteria for concentrated fruit & veg juices and free fatty acids. For the Slovenian logo, I believe also E% should be marked.
In general, it is unclear how the funnel model distinguishes between the reference base for an underlying nutrient profiling model and the reference base for the numerical information displayed (see UK MTL).
Furthermore, in my view it creates a wrong impression to say that Nutri-Score is limited to specific products when essentially all products subject to mandatory nutrition labelling could carry it. Perhaps the same applies to the Israeli system, for which I do not have an English version of the criteria.
Concerning the call for more research (specifically ll. 348-351) I doubt much more meaningful information can be obtained about individual features of FOP labelling schemes. Common sense has it that a labelling system that is easily spotted (size, contrast) and read (legibility) has a better chance of being paid attention to. The more intuitive the scheme is, the better its chances are for being understood and used. Accompanying education and awareness measures would then help optimise understanding and use. The 12-country study by Egnell et al. (2018) shows that there are schemes that work reasonably well across multiple sociocultural contexts. On these grounds, from a public health perspective it would seem prudent to now focus on organising a structured roll-out of such schemes and then monitor and evaluate the performance against suitable counterfactual controls. Research efforts should also go into understanding how much a given FOP scheme impacts on people's diets in real life and what this means for individual and public health (rightly noted in ll. 326-329). Unintended consequences, good or bad, should be taken note of in the process.
Minor comments
l. 16 + 55 – qualify statement that "all FOP labels currently existing in Europe and Israel" are being considered
ll. 22-24 – statement on implementation and evaluation is outside the scope of the study – suggest deletion
ll. 33-34 – sub-clause "addressing the policy response…" does not make sense to me. Suggest rewording that WHO recommends FOP labelling as a policy tool.
ll. 37-38 – more recent and FOP labelling-specific reviews should be cited here, such as Hawley et al 2012, Cecchini & Warin 2016, Hersey et al 2013. Probably also a good moment to include the latest WHO Europe report on FOP labelling.
l. 40 + 45 – "…used by FOP labelling systems" reads odd to me. Probably better "used in".
l. 43 – delete "their effectiveness in terms of"
ll. 60-62/Table 1 – use aspect descriptors also directly in the funnel model graphics
Table 1 – Explanation may be a better column label than Definition. Reference unit can also be per 100 or 1000 kcal, per 1 MJ etc. Since the table is meant to be generic, such information should be included. Purpose can also be to neutrally inform consumers about the nutritional contribution a product makes to the diet, which is different from identifying, and yet more different from actually choosing healthy foods. In Methodological approach, delete "of all" in line two of the definition. On tone of voice, an MTL may indeed convey a mixed message about a product's healthfulness, if reds and greens are combined and need to be traded off. However, the Nutri-Score provides a single summary indicator; hence the message is not mixed in my view but rather falls somewhere along the continuum from healthy to unhealthy. A native speaker might be able to help clarify this point (see also l. 135).
l. 66 – correct sentence to "…were used to complete…"
Figure 1 - What is the point of distinguishing between SFA+TFA on the one hand, and SFA and TFA on the other? Why is added sodium listed, but not added salt? Maybe it is worthwhile just distinguishing between added sodium/salt and total sodium/salt? The funnel model is not specific enough per se to inform detailed technical discussions on the matter anyway. The model would benefit from descriptors for the different sections/features, and what is the rationale for subdividing it into three main sections? Is there any hierarchy implied?
l. 88 – Supplementary figures are labelled S1 to S3, not A1 to A3.
l. 117 – suggest deleting "and received a national and EU approval in 2013" as the recent termination renders this irrelevant
l. 120-121 – Suggest rewording so it becomes clearer what the similarities and the differences between Choices and Keyhole are. Also, only a single qualifying nutrient considered in Choices, which should be made clearer.
Figure 3 - Are the Choices International criteria applied unaltered in any country? The map in Figure 3 wrongly suggests the logo is used in all countries.
l. 132 – The cited references seem not to provide any information on the status of Spain and Portugal regarding the Nutri-Score.
l. 145 – The Nutri-Score label always shows all 5 coloured boxes, but the product rating is indicated by the appropriate box being magnified. As for the threshold values, maybe better to say ranges or lower and upper bounds are defined for each of the five scores.
l. 148 – Statement incorrect as is. A generic Nutri-Score would allow for classification, but the score would not be in line with dietary recommendations; hence the adaptation for fats, cheeses, and beverages.
l. 172 + 182 – Abbreviations for SFA and TFA need to be introduced at first mention in l. 123 and also included in the figure legends
ll. 190-191 – the identification of the drivers behind specific FOP schemes is open to debate. See for example conflicting information with the statement on the Keyhole in ll. 93-94. Nutri-Score was developed by academics and then adopted/endorsed by the French Ministry of Health as the preferred voluntary FOP scheme to be implemented. Maybe clarify better what you mean by driver.
Table 2 – If you keep this table, then Nutri-Score coverage needs to be corrected to Specific products to be in line with the corresponding funnel model, or the funnel model corrected to say "all products" (see comment above).
l. 195 + 198 – "Of all mixed and negative FOP labels" reads odd given that you're only discussing three schemes.
l. 202 – Nutri-Score sets a generic cut-off of 1.2% for alcohol content
l. 210 – Considering the various adaptations, suggest qualifying the statement "all use one set of criteria for all food categories" as to be taken in a very broad sense
l. 216 – Nutri-Score initiated by academia, Keyhole by retailer, etc. Suggest to find a description that reflects the reality.
ll. 220-221 – In my view the paper is about describing and comparing "European" FOP labels using the funnel model as a tool
l. 234 – Need to cite solid and direct evidence for the statement "Previous research…".
l. 256 – suggest deleting "primarily aiming to help consumers identify healthier choices" but maybe qualifying "…only label studied here"
l. 259 – the Israeli system is not about summarising the overall healthiness of a product
ll. 272-273 – Also a strong argument against positive logos and any schemes that would (most likely) never reach (near) complete penetration, as consumers would need to know if a product is not eligible for or does not participate in the scheme
l. 279 – correct phrase as follows "compositions on the basis"
l. 294 – The various inaccuracies, if confirmed, would suggest external verification was not as successful as hoped.
l. 324 – Suggest referring to studies that tested label features (FLABEL, Ares et al. 2018, Malam et al. 2009)
l. 346 – Explain what you mean by "effectiveness of FOP labels", i.e. effective in what regard.
l. 357 – Would cite the latest WHO Europe report on FOP labels and the study by Kanter et al. 2018 in place of ref 37
Meanwhile, the authors need to clarify how their current manuscript goes beyond what has been reported in a previous study (http://www.sciencedomain.org/abstract/5399).
Author Response
Comments and Suggestions for Authors
The study by van der Bend and Lissner presents a comparison of the characteristics of different front-of-pack (FOP) labelling schemes from Europe and Israel, using a self-developed graphical model. While the study is timely given the ongoing debate in the EU, the proposed funnel model only allows a standardised description of FOP scheme characteristics; care should be taken not to venture too far beyond the consequentially limited scope.
Major comments
While the funnel model allows for a standardised recording of a range of FOP scheme characteristics, it is not clear from the manuscript what the true value of this assessment is, and who would be the main users of the information obtained. The Codex Alimentarius Commission has recently produced a global inventory of FOP schemes and their characteristics - what does your study add to the Codex database?
The global inventory of the Codex Alimentarius Commission that the reviewer is referring to indeed partially overlaps the items included in the Funnel Model. However, as far as we know this global inventory has not been published, and while it is very valuable as it contains an overview of FOP labels, it does not include a further discussion on the criteria or methodological approaches and challenges of the FOP systems. Also, it does not provide an easy-to use, compact and comprehensive tool to describe FOP systems at a glance, while this is one of the main values of our paper; the Funnel Model can easily be used by stakeholders with different backgrounds, i.e. not only by people already familiar in the field (i.e. users are clearly mentioned in lines 57-59). This paper therefore provides a starting point to a more thorough discussion on several methodological issues and criteria of different FOP labels, also by paying special attention to additional evaluation/implementation activities carried out by FOP labelling initiatives. These aspects are mostly neglected in articles on FOP labelling, while they are crucial for the successfulness of a FOP label in terms of consumer understanding, use, trustworthiness, etc. Finally, our paper has been developed in collaboration with the FOP labels involved, which we consider a valuable addition as well.
What is the reasoning for using the funnel shape? From a semantic angle, the funnel model may be read from top to bottom as a specific combination of criteria resulting in one specific FOP scheme. Obviously this is not the case, thus the authors may want to reconsider the shape of the graphic.
The Funnel Model was previously developed to make sure that the structure of FOP labels is more easily interpretable and can be used for presentation purposes by stakeholders with different backgrounds, using an appealing graphic model. It shows how different criteria, methodologies and other features together form the fundaments for a specific FOP labelling scheme. So, it does not necessarily describe the process of the development of a FOP label (also as there may not simply be one typical process for this), but rather the different features that together form a FOP label, in an easy-to-interpret manner.
Why is the model divided into three sections, and what links the elements within the middle and the bottom sections? Furthermore, what is the meaning of the part saying "Testing/Validation"? Currently it does not seem to contain any information?
We agree with the reviewer that the order or grouping of the different characteristics may seem be a bit arbitrary. We have slightly adjusted the Model, by naming the three separate parts (i.e. ‘Components’, ‘Methodology’, ‘Expression’) and moving ‘Purpose ( i.e. Help consumer/Reformulation)’ to the ‘Expression’ section, as from our perspective the purpose of a FOP label manifests itself in practice (see adjusted Funnel Model). See also added explanation below.
The testing/validation part refers to a general action initiated for all FOP labels that are to be implemented. During the testing/validation, the combination of characteristics that are unique to the FOP label are evaluated in practice. Once successfully tested, the FOP label may be implemented and additional implementation and evaluation activities may be carried out by the main driver (not captured by the Funnel Model). We have added an extra sentence on this in the method section, see below.
Lines 64-68: ‘A combination of these aspects together form the fundaments of a FOP labelling system, which is generally tested or validated before actual implementation. The aspects described in the Funnel Model are grouped into three different sections, i.e. the ‘Component’, ‘Methodology’ and ‘Expression’ section. The latter two refer to the methodology used, and how the label manifests itself in practice, respectively.’
There is no clear rationale for the inclusion or exclusion of specific FOP systems. The authors claim that "all FOP labels currently existing in Europe and Israel" were compared, yet:
there is no mention of the most widespread FOP scheme called Reference Intakes (RI);
The reviewer is correct in mentioning this. However, for this paper, we chose to focus on (semi-)directive FOP labelling systems, as these systems are most often subject of the FOP labelling debate. We have changed the text accordingly.
Lines 73-76: ‘This paper focuses on all (semi-)directive FOP labels currently in practice or in preparation in Europe, with a national implementation, i.e., systems used by specific retailers were not included in this comparison. We have focused on (semi)directive labels specifically, as these are primarily subject to the current FOP labelling debate.’
the Israeli FOP system is not yet in use;
The reviewer is mentioning this correctly. However, there already is a sentence about this in the method section, in which we explain the value of adding the Israeli Warning label, even though it has not yet been implemented (see lines 80-82).
In addition to this, we have changed wording ‘all FOP labels currently existing in Europe and Israel’ to: ‘All FOP labels currently in practice or in preparation in Europe’ throughout the whole text (lines 16, 55-56, 73-74, 265, 343, 415-416).
the inclusion of the Czech Vim co jim logo creates a bias towards Choices (the Croatian Healthy Living logo was also inspired by Choices);
The goal of our paper is to give an overview of all (semi-)directive FOP logos in European countries. We therefore believe that both Choices and Vim co Jim should be present in the overview for the sake of completeness, as we aim to focus on all FOP labels that are implemented on a national level. The Croatian logo indeed is inspired by Choices, and the Finnish Heart logo and Keyhole also come from the same source of inspiration. The value of this overview is that it clearly shows that there are a multitude of positive logos in Europe, that are similar in several ways, while they have interesting differences as well.
other FOP systems such as the traffic light schemes used by the retailers Eroski in Spain, Continente in Portugal, Rimi in Estonia are not considered.
The reviewer mentions correctly that there are additional traffic light systems used by specific retailers (e.g., Delhaize also uses such a system). However, the aim of our paper is to focus on schemes with a national implementation, not corporate-specific ones. The logos mentioned by the reviewer are in use by individual retailers, which does not fit in with the overall aim of this comparison. To make this clearer, we have added a sentence on this in the method section.
Lines 73-75: ‘This paper focuses on all (semi-)directive FOP labels currently in practice or in preparation in Europe, with a national implementation, i.e., systems used by specific retailers were not included in this comparison.’
In light of the authors' statement that no positive logo was included in the 12-country study by Egnell et al., efforts should be made to show a more comprehensive picture of different FOP scheme types in the present study. In my view, the Vim co jim logo should be removed from the analysis. Furthermore, it would seem prudent to give more space to the UK Multiple Traffic Light-RI hybrid scheme in the main text, including the corresponding funnel model.
We fully agree with the reviewer that adding the UK MTL scheme in the main text, including the Funnel Model, will be a valuable addition to our comparison. The main reason for not including it in our paper originally was because of the length of the article, however, based on the reviewer’s comment we decided to add an extra paragraph on the MTL, including the corresponding Funnel Model. We would like to refer to lines 174-193.
However, as mentioned earlier in our response to the reviewer, to our opinion it does not seem logical to leave the Vim co Jim logo out, as it has been implemented at national level in the Czech Republic, and therefore it should definitely be part of this European comparison.
Another issue with the funnel model is evidenced by the supplemental figure S3. Apart from the driver (governmental vs commercial), all other boxes are coloured in the same, suggesting the two systems are essentially the same. However, research suggests that consumers try to avoid MTL with lots of reds and remain rather indifferent to MTL without any reds. Consequently, seemingly identical systems may yield very different responses in real life, a fact not or only poorly reflected in the funnel model. It is also noteworthy here that the visuals used for the MTL and the ENL are the same in the funnel models, which I would strongly advise against. Better to take the MTL graphic from the official UK guidance at https://www.gov.uk/government/publications/front-of-pack-nutrition-labelling-guidance
The reviewer is right that the MTL and ENL come across rather similar when comparing the Funnel Models, while there are clear differences between the labels with respect to main driver and reference unit (i.e. the ENL was governed by industry and its expression is determined based on portion sizes). While these only seem to be minor differences when comparing the Funnel Models, in practice these differences have large implications. To make this clear, we have added a sentence about this in the discussion.
Lines 345-349: ‘Secondly, one should keep in mind that, when comparing Funnel Models of different FOP labelling systems, minor differences may still have large implications in practice. For example, when comparing the Funnel Models of the ENL and MTL, these systems seem to be almost similar,
while the use of portion sizes in combination with involvement of industry has led to criticism towards the ENL among several stakeholders.’
Also, we have changed the visuals of both the ENL and MTL and instead included the official UK image that the reviewer recommends (see ENL and MTL Funnel Models). We still used the same visual for both MTL and ENL, to show that they are actually similar in practice, while the differences mentioned above make clear that the methodology of the systems are rather distinctive.
The authors point out that each FOP scheme has its own unique features (l. 20) and at the same time they lump all positive logos together in Table 2, thus leaving the reader unclear as to which reference units and purposes apply to which scheme, or who is the driver behind which scheme. This lack of clarity then hampers understanding of the field "Components", where both options listed apply to all. Overall, Table 2 seems of little use to me – suggest deletion.
The main purpose of Table 2 is to provide insight into overall differences between FOP labels with a different ‘Tone of voice’ and ‘Directivity’, since these aspects are often key items in the international FOP labelling debate. Aspects such as specific reference units, drivers, or purposes are already specified explicitly in the Funnel Models themselves – therefore we do not consider it relevant to repeat specific features for each FOP label in this table. For example, Table 2 clearly shows that all positive labels use a category-specific approach while the mixed and negative labels use an across-the-board approach. Also, the main purpose of the MTL is to help consumers making healthier choices while the other systems focus on reformulation as well.
Since the present study is all about the application of the funnel model as a comparative tool, the various supposed errors and inaccuracies noted in the model graphics may also be considered a major issue. For example, according to my information the Choices logo has E% criteria for "All other" non-basic foods for SFA, TFA, and added sugars. The Croatian Healthy Living mark does not have criteria for added sodium but for added salt, and it includes disqualifying criteria for concentrated fruit & veg juices and free fatty acids. For the Slovenian logo, I believe also E% should be marked.
We thank the reviewer for noting this. For both the new 2018 Choices International Criteria and the Czech Vim co Jim, there indeed is an optional Energy% criterion, so we have colored Energy% as well.
The Croatian criteria again indeed don’t have added sodium criteria but instead maximum salt portion, as well as added salt (so not only added salt as the reviewer states above). We thank the reviewer for the useful suggestions regarding SFA + TFA and added and total salt/sodium boxes in the Funnel Model. We have removed the SFA + TFA box and created one box for added salt/sodium and one box for total salt/sodium (see also minor comments below). Furthermore, the reviewer is correct that the Croatian criteria include free fatty acids, so we have added a box for this in the Funnel Model. We have adjusted all of this in Supplementary Table B1 as well.
With respect to the Croatian criterion ‘not from concentrated fruit or vegetable juice’ for the ‘Fruit/vegetable nectars or similar products’ product category; we have not added a separate box for this in the Funnel Model, as this rather is viewed as part of the definition of fruit/vegetable juices and not as separate disqualifying component.
We have checked this with the representative from Slovenian logo, and the criteria indeed include Energy% as reference unit, but also per 100 kcal. They have not noted this during approving the Funnel Model unfortunately. So, we have now added a Reference unit box ‘100 kcal/KJ’ to all Funnel Models and highlighted this box and the Energy% box for Slovenia (see Funnel Models). We also changed this in Table 2.
In general, it is unclear how the funnel model distinguishes between the reference base for an underlying nutrient profiling model and the reference base for the numerical information displayed (see UK MTL).
The Reference Unit in the Funnel Model focusses on the reference units for the underlying nutrient profiling model or product criteria, as mentioned in Table 1. In the text that we added for MTL in the Results section, we clearly explain what reference units for the numerical information displayed on pack.
Lines 184-186: ‘On pack, reference bases are provided per 100g/ml only, per 100g/ml and per portion, or per portion only (applies only for ‘energy + 4’). When the latter is applied, energy must be provided per 100g/ml in addition to per portion.’
Furthermore, in my view it creates a wrong impression to say that Nutri-Score is limited to specific products when essentially all products subject to mandatory nutrition labelling could carry it. Perhaps the same applies to the Israeli system, for which I do not have an English version of the criteria.
We believe that it actually is valuable to make this distinction of all pre-packaged foods versus not all pre-packaged foods. For example, if we would want to distinguish between Choices/Keyhole/Croatian HL mark and the Nutriscore/Israeli Warning label: pre-packaged raw vegetables/fruits, raw meat, (sparkling) water, herbs, teas, etc., are eligible to carry the Choices/Keyhole/Croatian HL label, while the Israeli label and Nutriscore are generally not to be used on these foods (Nutriscore may be applied to these foods but only when they contain a nutrition declaration that complies with the INCO regulation). However, to make the difference seem less extreme, we have changed the wording ‘All products’ / ‘Specific products’ to ‘All products’ / ‘Not all products’, respectively (see Funnel Models).
Concerning the call for more research (specifically ll. 348-351) I doubt much more meaningful information can be obtained about individual features of FOP labelling schemes. Common sense has it that a labelling system that is easily spotted (size, contrast) and read (legibility) has a better chance of being paid attention to. The more intuitive the scheme is, the better its chances are for being understood and used. Accompanying education and awareness measures would then help optimise understanding and use. The 12-country study by Egnell et al. (2018) shows that there are schemes that work reasonably well across multiple sociocultural contexts. On these grounds, from a public health perspective it would seem prudent to now focus on organising a structured roll-out of such schemes and then monitor and evaluate the performance against suitable counterfactual controls. Research efforts should also go into understanding how much a given FOP scheme impacts on people's diets in real life and what this means for individual and public health (rightly noted in ll. 326-329). Unintended consequences, good or bad, should be taken note of in the process.
We would like to emphasize here the relevance to discuss the different methodological approaches or criteria underlying FOP labelling schemes, especially since so many studies have already been conducted on consumer understanding. We believe this is a major value of our paper. Therefore, to highlight this we added lines 399-401 (see below). Furthermore, the reviewer is only referring to the visual aspects of a FOP label (“Common sense has it that a labelling system that is easily spotted (size, contrast) and read (legibility) has a better chance of being paid attention to. The more intuitive the scheme is, the better its chances are for being understood and used.”), but in our discussion section we are mainly focusing on research on ‘…the rationale behind and application (of the criteria) of FOP labelling systems in practice, e.g. by exploring types of qualifying versus disqualifying components, scoring versus threshold methods, across-the-board versus food category specific approaches, nutrient level per 100g versus per serving units, or mandatory versus voluntary use’ (lines 405-409). The effectiveness of several of these methodologies should then indeed be studied more thoroughly (in different sociocultural settings), as mentioned throughout the discussion section. We added an extra sentence on this, see lines 409-412 (below).
Lines 399-401: ‘Importantly, to date many studies on FOP labelling have focused on consumer understanding, while only limited research has been conducted on the criteria or methodological approaches used in FOP labelling systems.’
Lines 409-412: ‘It is essential that each of these features are studied repeatedly, not only in relation to consumer understanding, but also product reformulation, applicability, and ease of use in practice, in different cultural settings. Results of such studies may provide fundaments for a FOP labelling system to be implemented in multiple settings.’
Minor comments
l. 16 + 55 – qualify statement that "all FOP labels currently existing in Europe and Israel" are being considered
As stated in our response on the major comments of the reviewer, we changed this or comparable sentences to ‘all FOP labels currently in practice or in preparation in Europe’ or to a slightly different version of this sentence throughout the whole article (lines 16, 55-56, 73-74, 265, 343, 415-416).
ll. 22-24 – statement on implementation and evaluation is outside the scope of the study – suggest deletion
We do not agree on deleting this section, as to our opinion it really adds to the characteristics of FOP labels presented in the Funnel Models; FOP labels are not simply objects ‘functioning’ on their own; the implementation and evaluation activities give the actual meaning to these FOP labels in practice, determining how, where, when and to what extend they are used by consumers. Therefore, we chose not to delete this part.
ll. 33-34 – sub-clause "addressing the policy response…" does not make sense to me. Suggest rewording that WHO recommends FOP labelling as a policy tool.
We changed the sentence according to the reviewer’s suggestion.
Lines 33-34: ‘The WHO recommends FOP labelling as a policy tool, to tackle the global epidemic of obesity and diet-related non-communicable diseases.’
ll. 37-38 – more recent and FOP labelling-specific reviews should be cited here, such as Hawley et al 2012, Cecchini & Warin 2016, Hersey et al 2013. Probably also a good moment to include the latest WHO Europe report on FOP labelling.
We thank the reviewer for these suggestions and have added the systematic review and meta-analysis of Cecchini and the latest WHO report on FOP labelling (line 36). We also highlighted the added references in yellow.
l. 40 + 45 – "…used by FOP labelling systems" reads odd to me. Probably better "used in".
We agree, we have changed this (lines 40 and 45).
l. 43 – delete "their effectiveness in terms of"
We agree, we have deleted this part of the sentence (line 43).
ll. 60-62/Table 1 – use aspect descriptors also directly in the funnel model graphics
To our opinion it is not necessary to include this in the Funnel Model as Table 1 clearly describes the categories for each aspect descriptor. Also, instead we have added between brackets (lines 61-62) that the aspect indicators in the text are mentioned in the order from top to bottom (Funnel Model).
Table 1 – Explanation may be a better column label than Definition. Reference unit can also be per 100 or 1000 kcal, per 1 MJ etc. Since the table is meant to be generic, such information should be included. Purpose can also be to neutrally inform consumers about the nutritional contribution a product makes to the diet, which is different from identifying, and yet more different from actually choosing healthy foods. In Methodological approach, delete "of all" in line two of the definition. On tone of voice, an MTL may indeed convey a mixed message about a product's healthfulness, if reds and greens are combined and need to be traded off. However, the Nutri-Score provides a single summary indicator; hence the message is not mixed in my view but rather falls somewhere along the continuum from healthy to unhealthy. A native speaker might be able to help clarify this point (see also l. 135).
We agree to and have adjusted all suggested changes of the first section, about Table 1. The comment about indicating the Nutriscore as a mixed label we have further clarified in the text about Nutriscore, i.e. by explaining that food products will be rated somewhere along the continuum from healthy to unhealthy; therefore for some foods the Nutriscore may be green and for other foods the Nutriscore may be light or dark orange. As this varies depending on the product, we cannot say it is either positive or negative, therefore it is considered mixed.
Lines 155-158: ‘Because Nutriscore provides a summary indicator for each food along the continuum from healthy to unhealthy, it is considered neither positive nor negative. Therefore, it is rather viewed as a mixed scheme.’
l. 66 – correct sentence to "…were used to complete…"
Agree, we have changed this accordingly (line 85).
Figure 1 - What is the point of distinguishing between SFA+TFA on the one hand, and SFA and TFA on the other? Why is added sodium listed, but not added salt? Maybe it is worthwhile just distinguishing between added sodium/salt and total sodium/salt? The funnel model is not specific enough per se to inform detailed technical discussions on the matter anyway. The model would benefit from descriptors for the different sections/features, and what is the rationale for subdividing it into three main sections? Is there any hierarchy implied?
The reviewer is right to mention this. We have now named the three sections in the Funnel Models to distinguish them better (see previous answer to the reviewer’s major comment. Also, based on the reviewer’s comments we have removed the SFA + TFA box and added a box for added sodium/salt and total sodium/salt (see Funnel Model).
l. 88 – Supplementary figures are labelled S1 to S3, not A1 to A3.
We agree, and have adjusted the labelling of the Supplementary Figures.
l. 117 – suggest deleting "and received a national and EU approval in 2013" as the recent termination renders this irrelevant
We actually believe it is relevant to mention that this label was notified once to make clear that the logo actually was recognized as a national FOP logo, even though it was terminated later. Therefore, we did not delete this sentence.
l. 120-121 – Suggest rewording so it becomes clearer what the similarities and the differences between Choices and Keyhole are. Also, only a single qualifying nutrient considered in Choices, which should be made clearer.
We have added a line specifying the type of foods that are covered by Choices and not by Keyhole (i.e. hedonic products). However, we don’t find it necessary to add that Choices only considers one qualifying nutrient, as this is already clearly mentioned in the text (line 138-139): ‘…i.e. minimum values for fiber, and maximum values for energy, SFA, TFA, sodium, total sugars and added sugars, respectively.’.
Lines 136-137: In contrast to Keyhole, the Choices criteria have been developed for all types of foods, including hedonic products such as snacks, sweets and soft drinks.
Figure 3 - Are the Choices International criteria applied unaltered in any country? The map in Figure 3 wrongly suggests the logo is used in all countries.
The only country that has accepted the Choices criteria unaltered is Argentina. The other countries have tailored it to the national situation (Czech Rep, Croatia, Nigeria, China, Zambia) or developed their own criteria using the Choices criteria as an example (Malaysia, Thailand, Brunei). Choices claims a global coverage because the international criteria are based on a global analysis of nutrition recommendations, dietary guidelines, food cultures and food composition databases. For example, the reference database for the current criteria revision is made up from product data from UK, US, Australia, New Zealand, China, India, and used criteria sets of different logo programs from Asia, Africa and Europe. This global coverage also accounts for the ENL. To make clear that the map does not show actual presence but reference, we have added a line in the text about Choices.
Lines 124-126: ‘Because the Choices International criteria are based on a global analysis on nutrition recommendations, dietary guidelines, and food composition databases, and therefore aims for global coverage, the Funnel Model map is fully colored. The same applies to the Evolved Nutrition Label.’
l. 132 – The cited references seem not to provide any information on the status of Spain and Portugal regarding the Nutri-Score.
We were not aware of this, and have now added the correct references for Spain and Portugal (line 149-150). See also the highlighted references in yellow.
l. 145 – The Nutri-Score label always shows all 5 coloured boxes, but the product rating is indicated by the appropriate box being magnified. As for the threshold values, maybe better to say ranges or lower and upper bounds are defined for each of the five scores.
We have changed this according to the reviewer’s comments.
Lines 165-166: ‘Which box (A-E) will be magnified depends on specific lower and upper bounds that are defined for each of the five boxes.’
l. 148 – Statement incorrect as is. A generic Nutri-Score would allow for classification, but the score would not be in line with dietary recommendations; hence the adaptation for fats, cheeses, and beverages.
We agree with the reviewer, and have changed this sentence according to the reviewer’s comments.
Lines 166-169: ‘The Nutriscore is based on one set of criteria for all pre-packaged foods, although criteria modifications have been made specifically for cheeses, fats and non-alcoholic drinks, because the score of these products would not be in line with dietary recommendations.’
l. 172 + 182 – Abbreviations for SFA and TFA need to be introduced at first mention in l. 123 and also included in the figure legends
We have adjusted this accordingly (see lines 139, 221, 225). Furthermore, we have altered the Funnel Models using the full terms, i.e. ‘saturated fat’ and ‘trans fat’. For consistency, we did this also for MUFA, PUFA, DHA and EPA (see Funnel Models).
ll. 190-191 – the identification of the drivers behind specific FOP schemes is open to debate. See for example conflicting information with the statement on the Keyhole in ll. 93-94. Nutri-Score was developed by academics and then adopted/endorsed by the French Ministry of Health as the preferred voluntary FOP scheme to be implemented. Maybe clarify better what you mean by driver.
We would like to clarify here that main driver is the actor currently responsible for implementation of the FOP label in practice. We have added this to Table 1.
Explanation ‘Driver’ Table 1: ‘This refers to the driving force behind a FOP label at the time of the writing of this article; a driver may be governmental, commercial, or be part of an NGO.’
Table 2 – If you keep this table, then Nutri-Score coverage needs to be corrected to Specific products to be in line with the corresponding funnel model, or the funnel model corrected to say "all products" (see comment above).
This is correctly mentioned by the reviewer. However, as stated earlier in our responses to the reviewer’s comments, Nutriscore does not cover all pre-packaged foods, therefore we changed coverage to ‘Not all products’, see Table 2.
l. 195 + 198 – "Of all mixed and negative FOP labels" reads odd given that you're only discussing three schemes.
We agree with the reviewer and have adjusted these sentences.
Lines 236-238: ‘Similar to the positive labels, the mixed and negative FOP labels include disqualifying components in their criteria. However, they don’t include any qualifying components, except for Nutriscore (Supplementary Table B2).’
l. 202 – Nutri-Score sets a generic cut-off of 1.2% for alcohol content.
In the Nutriscore criteria document we indeed found that Nutriscore does not apply to alcoholic drinks containing more than 1.2% alcohol, however, alcoholic drinks are not considered by any of the FOP labels anyway (See Table 1 for specification of Coverage). Besides alcoholic drinks, other specific foods such as infant formula and food supplements are also not covered by the FOP labelling systems, so we don’t consider them anyway here.
l. 210 – Considering the various adaptations, suggest qualifying the statement "all use one set of criteria for all food categories" as to be taken in a very broad sense
We agree with the reviewer and have adjusted this sentence.
Lines 250-254: ‘The most important difference between the positive labels and the mixed or negative labels is that the positive FOP labels all use a category-specific approach, i.e. applying different criteria for different food categories, while the mixed and negative FOP labels generally use one set of criteria for all food categories, with sometimes a few exceptions regarding specific food categories, e.g. see Nutriscore.’
l. 216 – Nutri-Score initiated by academia, Keyhole by retailer, etc. Suggest to find a description that reflects the reality.
We fully agree. We have already changed the explanation of main driver in Table 1, but will additionally change ‘initiated by’ to ‘governed by’ in all cases (lines 232, 260, 261).
ll. 220-221 – In my view the paper is about describing and comparing "European" FOP labels using the funnel model as a tool
We agree and have changed this.
Lines 264-266: ‘This paper describes all FOP labels currently in practice or in preparation in Europe using the Funnel Model as a tool.’
l. 234 – Need to cite solid and direct evidence for the statement "Previous research…".
Reference is already present in text (i.e. after the next sentence). I will cite this reference again after the sentence mentioned by the reviewer, including a paper by the European Food Safety Authority (see line 281). See new reference highlighted in yellow.
l. 256 – suggest deleting "primarily aiming to help consumers identify healthier choices" but maybe qualifying "…only label studied here"
We agree and have adjusted the text accordingly.
Lines 300-301: ‘The MTL label is the only label studied here not specifically stimulating product reformulation.’
l. 259 – the Israeli system is not about summarising the overall healthiness of a product
We agree. We have changed the sentence as suggested by the reviewer.
Lines 301-304: ‘Additionally, the MTL label is distinctive from the other FOP labels in that it is semi-directive, combining colors with nutrient values; the other FOP labels are all directive, providing a summary indicator to describe the nutritional quality or ‘healthiness’ of a food product.’
ll. 272-273 – Also a strong argument against positive logos and any schemes that would (most likely) never reach (near) complete penetration, as consumers would need to know if a product is not eligible for or does not participate in the scheme
Agree, we added the last part of the reviewer’s sentence to the text.
Lines 319-320: ‘Ideally, consumers would need to know whether a product is either not eligible for or does not participate in the FOP labelling scheme.’
l. 279 – correct phrase as follows "compositions on the basis"
We have adjusted the sentence.
Line 324-326: ‘A ‘per serving’ approach can introduce several difficulties; not only is it challenging to compare products’ compositions on the basis of their serving size, but also serving sizes and consumption patterns are individual matters which cannot easily be standardized.’
l. 294 – The various inaccuracies, if confirmed, would suggest external verification was not as successful as hoped
To our opinion, the Funnel Model is still reliable even though some minor issues were overlooked. We do believe that the extra verification by the main drivers is useful, as it was an extra check to verify our initial input, and small mistakes are often part of such a process. The Funnel Model should now be fully corrected thanks to the observations of the reviewer.
l. 324 – Suggest referring to studies that tested label features (FLABEL, Ares et al. 2018, Malam et al. 2009)
We thank the reviewer for these reference suggestions and have added Malam et al (2009) (line 375). See new reference also highlighted in yellow.
l. 346 – Explain what you mean by "effectiveness of FOP labels", i.e. effective in what regard.
Not applicable anymore; this sentence was removed as a result of previous comments of the reviewer.
l. 357 – Would cite the latest WHO Europe report on FOP labels and the study by Kanter et al. 2018 in place of ref 37
We thank the reviewer for these suggestions, and have changed the references accordingly (line 425). See new references also highlighted in yellow.
Meanwhile, the authors need to clarify how their current manuscript goes beyond what has been reported in a previous study (http://www.sciencedomain.org/abstract/5399).
Originally, the 2014 Funnel Model was created to be used as a tool for updating information regarding FOP labels. Our paper builds on the 2014 paper by re-using and re-shaping the Funnel Model to update the overview of FOP labels in Europe, in collaboration with the FOP labelling initiatives (which we consider a novel aspect as well). Our paper goes beyond the 2014 paper as it more elaborately describes the methodological approaches and criteria used by each scheme, while adding implications of these in terms of effectiveness and further research. We also believe that the overall clustering of positive, mixed and negative FOP labels and their comparison provides a comprehensive overview in the European FOP labelling debate. In addition, this paper includes implementation and evaluation in practice, which makes the overview complete. See added lines below.
Lines 401-405: ‘Our paper is a highly valuable addition to the 2014 Funnel Model paper, contributing to the FOP labelling debate by elaborately describing and discussing the underlying methodologies used in positive, mixed and negative FOP labelling systems as well as their potential implications in practice, while acknowledging the importance of additional evaluation and implementation activities carried out by FOP labelling initiatives.’